# Super-Resolution Microscopy Analysis of Hepatitis B Viral cccDNA and Host Factors

**DOI:** 10.3390/v15051178

**Published:** 2023-05-16

**Authors:** Phuong Thi Bich Doan, Kouki Nio, Tetsuro Shimakami, Kazuyuki Kuroki, Ying-Yi Li, Saiho Sugimoto, Hideo Takayama, Hikari Okada, Shuichi Kaneko, Masao Honda, Taro Yamashita

**Affiliations:** Department of Gastroenterology, Graduate School of Medicine, Kanazawa University, 13-1 Takara-Machi, Kanazawa 920-8641, Japan; doanthphuong@gmail.com (P.T.B.D.);

**Keywords:** HBV, RNA Pol II, H3K4me3, fluorescence in in situ hybridization FISH, transcription factors

## Abstract

Infection with hepatitis B virus (HBV) cannot be cured completely because of the persistence of covalently closed circular DNA (cccDNA). We previously found that the host gene dedicator of cytokinesis 11 (DOCK11) was required for HBV persistence. In this study, we further investigated the mechanism that links DOCK11 to other host genes in the regulation of cccDNA transcription. cccDNA levels were determined by quantitative real-time polymerase chain reaction (qPCR) and fluorescence in situ hybridization (FISH) in stable HBV-producing cell lines and HBV-infected PXB-cells®. Interactions between DOCK11 and other host genes were identified by super-resolution microscopy, immunoblotting, and chromatin immunoprecipitation. FISH facilitated the subcellular localization of key HBV nucleic acids. Interestingly, although DOCK11 partially colocalized with histone proteins, such as H3K4me3 and H3K27me3, and nonhistone proteins, such as RNA Pol II, it played limited roles in histone modification and RNA transcription. DOCK11 was functionally involved in regulating the subnuclear distribution of host factors and/or cccDNA, resulting in an increase in cccDNA closely located to H3K4me3 and RNA Pol II for activating cccDNA transcription. Thus, it was suggested that the association of cccDNA-bound Pol II and H3K4me3 required the assistance of DOCK11. DOCK11 facilitated the association of cccDNA with H3K4me3 and RNA Pol II.

## 1. Introduction

Hepatitis B virus (HBV) is responsible for causing a potentially life-threatening chronic liver infection in 296 million people worldwide [1], which can lead to serious liver disease, including hepatocellular carcinoma and cirrhosis.

HBV belongs to the Hepadnaviridae family of viruses, and its small 3.2-kb genome consists of partially double-stranded DNA. The virological persistence of HBV is the result of its covalently closed circular DNA (cccDNA), which can elude treatment with nucleos(t)ide analogs. cccDNA minichromosomes in the nucleus enable recurrent and persistent HBV replication, suggesting the value of clarifying the molecular details of cccDNA maintenance. 

Because of its supercoiled nature, HBV DNA in the nucleus of infected hepatocytes exists as individual minichromosomes with cellular chromatin-like structures [2,3], which are responsible for the production of new viral DNA genomes through reverse transcription to pregenomic RNA (pgRNA) and subgenomic RNAs. The nucleosomes along viral minichromosomes are sequence-specifically positioned [4], suggesting that, similar to host cellular chromatin, the dynamic changes in the transcription of cccDNA and its stability are regulated by nucleopositioning and histone modification [3,5]. However, it remains unclear how host factors and HBV cccDNA interact on a single-molecule scale.

Genome-wide studies have revealed two mechanisms for histone modification; one involving the disruption of chromatin contacts, and the other involving the recruitment of nonhistone proteins to chromatin. The loss of H3K4me3 decreases transcriptional activity and the recruitment of heterochromatin protein 1 (HP1) and leads to a lack of histone deposition of H3K27me3, which is frequently linked to the inhibition of cccDNA transcription [5,6,7,8]. There is increasing evidence that the transcriptional activity of cccDNA is predominantly determined by host RNA polymerase II (RNA Pol II) machinery [9]. In addition, through single-cell transcriptome analysis using a newly established cell line called KM, which maintains long-term HBV infection, originating from a HCC patient, and xenotransplanted into NOD/SCID mice, in which we could detect cccDNA and HBV DNA, as well as HBV-derived transcripts, we recently identified the dedicator of cytokinesis 11 (DOCK11), which regulates cytokinesis by acting through guanine exchange factor (GEF) activity as a host factor associated with HBV maintenance [10]. Moreover, gene silencing by the short hairpin RNA (shRNA) knockdown of DOCK11 significantly reduced the level of HBV cccDNA transcription in the HBV-inducible cell line HepAD38. This suggests that DOCK11 might participate in the regulation of HBV cccDNA maintenance.

In this study, we combined the ViewRNA™ technology, which involves RNA fluorescence in situ hybridization (FISH), a powerful technique that adopts both FISH and sequential branched DNA amplification to visualize the major HBV nucleic acids at a single-molecule level, and with chromatin immunoprecipitation (ChIP) to clarify the interaction of the host gene DOCK11 with HBV cccDNA, the histone proteins H3K4me3 and H3K27me3, and the nonhistone proteins HP1α and RNA Pol II. 

## 2. Materials and Methods

### 2.1. Cell Lines

HEK 293T, HepAD38, and HepG2.2.15 cells were obtained from the American Type Culture Collection (Manassas, VA, USA) or the Japanese Collection of Research Bioresources Cell Bank (Osaka, Japan). HEK 293T and HepG2.2.15 cells were cultured in DMEM (Nacalai Tesque, Inc., Kyoto, Japan). HepAD38 were maintained in DMEM/F-12 with GlutaMAX™ supplement (Thermo Fisher Scientific, Waltham, MA, USA), 10% fetal bovine serum, 100 U/mL penicillin, 2 mM l-glutamine, and 100 µg/mL streptomycin. In addition, 400 µg/mL G418 and 10 nM HEPES buffer solution was added to the medium for the HepAD38 cells.

### 2.2. HBV Infection of Primary Human Hepatocyte

Primary human hepatocytes (PXB-cells^®^, PhoenixBio Co., Ltd., Hiroshima, Japan) were cultured in 12-well plates (2.1 × 10^5^ cells/cm^2^) in specific medium (#PPC-M200; PhoenixBio) without passaging and were infected overnight with HBV particles (the serum collected from HBV-infected PXB mice, genotype C; #PPC-BC, PhoenixBio) at 5 genome equivalents (GEq)/cell in the presence of 4% PEG 8000 and 2% DMSO. After 24 h, the cells were washed three times with phosphate-buffered saline (PBS). On Days 7, 14, and 21 the cells were collected to extract RNA and DNA and passaged to an 8-well chamber for hybridization.

### 2.3. Knockdown of DOCK11 Using shRNA

DOCK11-target shRNA and a non-target shRNA control (Sigma-Aldrich, St. Louis, MO, USA) were used for the knockdown of the DOCK11 gene in HepAD38 cells, as previously described, but with slight modifications [10]. Briefly, shRNA was cloned into a lentivirus-based pLKO.1-puro shRNA expression vector. On Day 0, the lentiviral particle supernatant was added to the cells, and they were incubated overnight at 37 °C. On Day 1, the cells were washed twice with PBS, and medium for puromycin selection was added. From Days 3–6, the culture medium was replaced as appropriate. On Day 6, the cells were collected for RNA and DNA isolation or passaged to an 8-well chamber for hybridization.

### 2.4. Overexpression of DOCK11

Halo-tagged DOCK11 cDNA plasmid (Kazusa DNA Research Institute, Chiba, Japan) and pFN21A plasmid (empty vector control) were transfected into HepAD38 cells using transfection reagent (Lipofectamine 3000; Thermo Fisher Scientific), following the manufacturer’s protocol. At 4–6 h post-transfection, the cells were washed with PBS and then cultured in DMEM with 10% fetal bovine serum.

### 2.5. Quantitative Real-Time PCR

Total RNA and HBV DNA were obtained using an ISOPIN Cell and Tissue RNA Kit (Nippon Gene Co., Ltd., Tokyo, Japan) and a DNeasy Blood & Tissue Kit (Qiagen, Hilden, Germany), respectively, following the manufacturer’s protocols. To isolate cccDNA, the extracted DNA was treated with T5 exonuclease (NEB #M0663; New England Biolabs, Boston, MA, USA) for 30 min at 37 °C, followed by 5 min at 95 °C for inactivation. Both cccDNA and HBV DNA were measured three times each using a real-time PCR system (QuantStudio 12K Flex; Thermo Fisher Scientific) with specific probes. The primer sets for the cccDNA and HBV DNA are shown in Appendix A.

### 2.6. Western Blotting

RIPA buffer (cat. no. #20-188; Merck Millipore, Burlington, MA, USA) was used to extract the cell lysates, after which the protein samples were separated using 10% sodium dodecyl sulfate–polyacrylamide gel electrophoresis, transferred onto polyvinylidene fluoride membranes, blocked with 5% skim milk, and incubated with the primary antibody overnight at 4 °C. The next day, incubation with secondary antibodies (1:2000; cat. no. #7074; Cell Signaling Technology, Danvers, MA, USA) was carried out at room temperature for 1 h. Finally, enhanced chemiluminescence detection reagents (Amersham Biosciences Corp., Piscataway, NJ, USA) were used to visualize the immune complexes, following the manufacturer’s protocols.

### 2.7. Chromatin Immunoprecipitation

One week later, ChIP experiments were performed in DOCK11-knockdown and control cells, using a ChIP kit (SimpleChIP^®^ Enzymatic Chromatin IP Kit (Magnetic Beads), #9003; Cell Signaling Technology), following the manufacturer’s protocol. Briefly, protein–DNA complexes were cross-linked using 1% formaldehyde and then quenched with 0.125 M glycine. To prepare the nuclear extract, the cells were lysed in lysis buffer containing protease inhibitor. The samples were sonicated to reduce the chromatin to an average length of about 1 kb, and they were ultracentrifuged; the supernatant was collected and diluted 1:10 with dilution buffer, and the appropriate antibodies were added. Chromatin was immunoprecipitated overnight at 4 °C, using 10 µg of the antibodies listed in Appendix A. Normal rabbit IgG antibody was used as the negative control, while histone H3 antibody was used as the positive control. Then, the immune complexes were incubated with protein A/protein G agarose beads at 4 °C, washed, and eluted in 1× ChIP buffer. Finally, using RLP30- and cccDNA-specific primers, the immunoprecipitated DNA was quantified by qPCR.

### 2.8. FISH of Viral RNA, Relaxed Circular DNA, and cccDNA

Hybridization and amplification were carried out using an RNA assay kit (ViewRNA™ Cell Plus Assay Kit, #88-19000-99; Thermo Fisher Scientific) with an HBV probe set, following the manufacturer’s protocol. Based on the report by Zhang et al. [11], the cells were cultured in 10 cm wide dishes and seeded on collagen-coated 8-well chamber slides. The chambers were fixed using the solution contained in the ViewRNA Cell Plus kit before antibody staining and re-fixation for 1 h. The cells were pretreated with RNase A (#EN0531; Thermo Fisher Scientific), Rnase H (#EN0201; Thermo Fisher Scientific), Dnase I (#EN0521; Thermo Fisher Scientific), or T5 exonuclease (NEB #M0663; New England Biolabs), and then re-fixed with 3.7% formaldehyde for 5 min. The probe sets were diluted 100-fold in the probe set diluent buffer supplied with the kit, and then the pre-amplifier and amplifier steps were carried out to enhance specificity and individual signals for target detection. Finally, probe labeling and nuclear staining (Hoechst 33342, #H3570; Invitrogen, Carlsbad, CA) were performed.

### 2.9. Immunofluorescence

HepAD38 cells were seeded in 8-well chambers (#5232-008; Iwaki & Co., Ltd., Tokyo, Japan), fixed with 4% formaldehyde in PBS at room temperature for 15 min, and permeabilized with 0.1% Triton X-100. The cells were then blocked with 1% BSA solution at room temperature for 1 h, incubated overnight at 4 °C with primary antibodies (diluted 1:1000), rinsed five times with PBS for 2 min each, and incubated with fluorescent secondary antibodies for 1 h. Nuclear staining was performed with Hoechst 33,342 for 5 min at room temperature. The presence of recombinant proteins in hepatocytes was checked using a confocal microscope system (Dragonfly CR-DFLY-301; Oxford Instruments, Abingdon, UK), and the images were analyzed using Fusion ver. 2.1.0.80 (Andor Technology, Belfast, UK).

### 2.10. Statistical Analysis

Data are presented as means ± the standard deviations. All experimental procedures were carried out three or more times. The data were evaluated using two-tailed unpaired Student’s *t*-test or one-way analysis of variance. Data analysis was performed using GraphPad Prism 8 software (GraphPad Software, San Diego, CA, USA), with *p* < 0.05 considered to indicate statistical significance.

## 3. Results

### 3.1. FISH with Branched DNA Amplification Facilitates the Subcellular Localization of Key HBV Nucleic Acids in HBV-Inducible Hepatocytes by Super-Resolution Confocal Microscopy

The observation of HBV DNA, RNAs, and cccDNA at single-molecule resolution facilitate provided insights into the interaction of key HBV nucleic acids with host factors. Here, we employed FISH with the branched DNA amplification method for the direct visualization of HBV DNA, RNAs, and cccDNA with fluorescent probes in order to identify proteins potentially associated with HBV (Figure 1A) [12]. We used two probe sets that could target HBV DNA, pgRNA, and cccDNA in combination with various DNases and RNases. 

The two different probes complementarily hybridized to different sites of the viral genome, depending on the target, probe set 1 (nt 2959–837, based on GenBank AB675675, # VF1-13851, Thermo Fisher Scientific), can bind to the minus strand of the HBV genome, and thereby detect signals only from HBV DNA. Probe set 2 (nt 1091–1557, # VPZTD39, Thermo Fisher Scientific) enables the identification of cccDNA because the viral RNAs and integrated HBV DNA are degraded by RNases and T5 exonuclease treatment, respectively.

The specificity of these probes was confirmed by a hybridization reaction in the HepAD38 cell line, which can induce HBV particles (genotype D), followed by pretreatment with DNase and/or RNase. Hybridization with probe set 1 yielded a strong signal for HBV DNA in both the nucleus and cytoplasm. After RNase A treatment, most of the signal remained intact, whereas DNase I treatment removed most of this signal (Figure 1C–D), suggesting that probe set 1 specifically bound to HBV DNA and not HBV RNA. When cells were incubated with RNase A, Rnase H, and T5 exonuclease and hybridized with probe set 2, only a few dots accumulated in the nucleus (Figure 1F), suggesting that probe set 2 specifically binds to HBV cccDNA in the cells. Pretreatment with DNase I alone resulted in the detection of spotty cytoplasmic signals, most likely corresponding to HBV RNAs (Figure 1G). RNase A combined with DNase I pretreatment before hybridization erased the signals on the slides (Figure 1H). Taken together, these data confirmed that FISH with the branched DNA amplification method with ViewRNA probes successfully permitted the visualization of HBV nucleic acids in HepAD38 cells.

### 3.2. Detection of HBV Nucleic Acids in HBV-Replication Cell Lines

HepG2.2.15 and HepAD38 were used to evaluate the sensitivity of ViewRNA ISH technology in the detection of HBV nucleic acids. Although both HepG2.2.15 and HepAD38 cells can secrete complete HBV particles, their production in HepAD38 cells depends on tetracycline promoter activity. Accordingly, doxycycline treatment suppresses HBV replication and particle production in HepAD38 cells, and the amount of HBV cccDNA can plateau 1 week after the removal of doxycycline from the culture medium [13]. The FISH signals of pgRNA, HBV DNA, and cccDNA were barely detectable after doxycycline treatment (Figure 2A), whereas HepAD38 cells without doxycycline treatment showed robust accumulation of the HBV nucleic acids (Figure 2B). Similarly, high levels of pgRNA, cccDNA, and HBV DNA were detected in HepG2.2.15 cells after 1 week (Figure 2C), suggesting that HBV nucleic acid amounts can be measured in a highly sensitive manner by ViewRNA technology.

To directly determine the numbers of cccDNA in the nucleus, FISH signals detected in 200 nuclei of HepG2.2.15 and HepAD38 cells were counted. The control without induction and the induced HepG2.2.15 and HepAD38 cells exhibited an average of 1.51, 5.59, and 3.9 FISH signals per nucleus, respectively (Table 1 and Figure 2D–F). Without doxycycline, the average and median FISH signal counts of HepAD38 increased around 4-fold compared with the control. HBV replication was verified using qPCR (Figure 2G–I).

### 3.3. Detection of HBV Nucleic Acids in HBV-Infected PXB-Cells

To evaluate the kinetics of the levels of HBV RNA, HBV DNA, and cccDNA in hepatocytes after HBV infection, PXB-cells were infected with HBV particles at 5 GEq/cell. Then, the amount of HBV nucleic acid was measured using ViewRNA probes specific to HBV RNA, HBV DNA, and cccDNA. HBV nucleic acids increased gradually after HBV infection and did not reach a plateau by 3 weeks (Figure 3A–C). Three weeks post-infection, HBV DNA presented predominantly in the cytoplasm, and HBV RNA mainly localized in the nucleus. It also may be due to the secretion of HBV DNA into the cytoplasm and preparation for recycling that cccDNA continued to accumulate in the nucleus and served as a temple to the synthesis of HBV RNA. Consistently, PCR results from PXB-cells analyzed 1, 2, and 3 weeks after HBV infection showed a similar tendency (Figure 3D–F). Collectively, our data suggested that branched DNA amplification technology, which involves the linear amplification of nucleic acids with specific probes, can fully evaluate the amount of HBV nucleic acids without exponential amplification methods such as PCR, thereby facilitating the more precise quantitation of HBV DNA, HBV RNA, and cccDNA with spatial information. 

### 3.4. Determination of DOCK11 Partners for Promoting cccDNA Replication

We recently established KM cells (also called HC1 [10]), a hepatocellular carcinoma (HCC)-derived cell line in which HBV DNA, HBV RNA, and cccDNA could be detected in a small subset of cells. Using KM cells and single-cell transcriptome analysis, we found that the host factor DOCK11 is pivotal for maintaining HBV DNA and cccDNA [10]. Consistently, the expression level of DOCK11 was significantly increased at both the mRNA and protein levels in HBV-infected cells compared with non-infected cells (Appendix A). We also analyzed a publicly available dataset on human liver tissue (GEO: GSE83148) and found that the expression of DOCK11 in hepatocytes was influenced by HBV infection and was increased in patients with chronic hepatitis B (Appendix A). Although DOCK11 protein is involved in intracellular signaling networks for GEFs for Cdc42 and Rac [14], it was not clear how DOCK11 maintains HBV replication and cccDNA maintenance.

Therefore, we evaluated the subcellular localization of DOCK11 by super-resolution microscopy, which enables the visualization of target protein at a resolution of at least 50 nm [15,16]. Surprisingly, we found DOCK11 in both the nucleus and cytoplasm in HepAD38 cells (Appendix A). Furthermore, we found the clear perinuclear localization of DOCK11 adjacent to lamin A/C (Appendix A). In general, active RNA transcription accompanies the open chromatin status marked with H3K4me3, mainly located in the central nuclear area, whereas inactive RNA transcription accompanies H3K27me3 and HP1 protein closed chromatin marks, mainly located in the perinuclear area. Host RNA Pol II has been reported to be in charge of cccDNA transcription [9]. While H3K4me3 is recognized as an activator of cccDNA transcription, H3K27me3 is associated with transcriptional silencing [17]. Furthermore, the recruitment of HP1 is frequently linked to the inhibition of cccDNA transcription [8]. Therefore, we attempted to determine whether DOCK11 colocalizes with RNA Pol II, H3K4me3, HP1 and H3K27me3.

DOCK11 partially colocalized with H3K4me3 (Figure 4A) and RNA Pol II (Figure 4B) at the center of the nucleus, whereas DOCK11 colocalized mainly with H3K27me3 (Figure 4D), adjacent to the nuclear membrane. HP1α was completely not complementary to DOCK11 (Figure 4C). To investigate whether DOCK11 physically interacts with these host factors, we performed co-immunoprecipitation using HepAD38 cells overexpressing DOCK11 (Figure 4E and Appendix A). We found no direct interaction with DOCK11 and these host factors (Figure 4F), suggesting limited roles for DOCK11 in histone modification and RNA transcription.

### 3.5. Localization of cccDNA with DOCK11 and Histone Markers Analyzed by Dragonfly Confocal Microscopy

HBV cccDNA is assembled with both histones and nonhistones to form what is known as a minichromosome. We used ViewRNA technology to determine the spatial and quantitative correlation of the cccDNA position with H3K4me3, H3K27me3, RNA Pol II, and HP1α. Interestingly, the cccDNA structure itself did not colocalize with DOCK11, but it did partially colocalize with H3K4me3, H3K27me3, or RNA Pol II (Figure 5A–E). These data are consistent with a recent model indicating that the transcription of HBV RNA from cccDNA depends on the open/closed chromatin status regulated by host factors [18]. The enrichment of the euchromatin markers H3K4me3 and RNA Pol II colocalized with cccDNA suggested that cccDNA is actively transcribed, whereas the colocalization of cccDNA with the heterochromatin markers H3K27me3 and HP1α suggested that cccDNA is inactive. To uncover the role of DOCK11 in the spatial distribution regulation of these markers in terms of cccDNA maintenance and replication, we lentivirally knocked down DOCK11 expression (Figure 5F) and found that both HBV DNA and cccDNA were decreased by DOCK11 knockdown (Figure 5G–H). We then evaluated the spatial distribution changes of cccDNA colocalized with open/closed chromatin markers, finding that DOCK11 knockdown resulted in a reduction in cccDNA colocalized with H3K4me3 and RNA Pol II, whereas no substantial changes were observed in the cccDNA colocalized with H3K27me3 and HP1α (Figure 5I–L). These data suggested that DOCK11, a GEF that regulates cytokinesis in cytoplasm, might be involved in regulating the subnuclear distribution of host factors and/or cccDNA, which might result in an increase in the cccDNA located near H3K4me3 and RNA Pol II to activate the transcription of HBV RNA from cccDNA.

### 3.6. DOCK11 Knockdown Decreases cccDNA-Associated H3K4me3 and Pol II C-Terminal Domain

The above findings encouraged us to investigate how DOCK11 regulates HBV cccDNA transcription. Therefore, we performed cccDNA ChIP-qPCR to check whether DOCK11 depletion affected the ability of host H3K4me3, H3K27me3, RNA Pol II, and HP1α to bind to the cccDNA minichromosome. The housekeeping gene RLP30 was used as a control marker for active transcription because it is highly expressed in HepAD38 cells. Confirming our expectations, DOCK11 knockdown markedly reduced the enrichment of H3K4me3 and RNA Pol II on cccDNA (Figure 6B). However, consistent with the binding preference for cccDNA, the binding levels of H3K27me3 and HP1α were relatively stable (Figure 6B). Taken together, these results suggested that cccDNA-bound RNA Pol II and H3K4me3 can be used as active markers of cccDNA transcription and that their association with cccDNA requires the involvement of DOCK11. 

## 4. Discussion

Antiviral treatments, such as nucleos(t)ide analogs, have been developed to reduce the viral loads of HBV. However, there is still a risk of hepatocarcinogenesis and HBV reactivation because of the sustained presence of nuclear cccDNA in hepatocytes. Accordingly, the development of a therapy that can eliminate cccDNA is of prime importance for eradicating HBV diseases. To do so, the elucidation of the dynamics and regulatory mechanisms of cccDNA are critical. Thus, in this study, we developed a method to more clearly visualize cccDNA and its nuclear binding molecules.

In HepAD38, HepG2.2.15, and HBV-infected PXB-cells, FISH was able to detect but could not distinguish the various forms of HBV DNA, namely relaxed circular DNA, double-stranded linear DNA, and single-stranded DNA (Figure 2B–C) [19,20]. For this reason, the HBV DNA was distributed in both the nucleus and cytoplasm. A previous study showed that one possible limitation of the FISH methodology is its lack of specificity in cccDNA determination [20] because the use of plasmids CMV-ayw for HBV probe preparation resulted in the detection of all HBV nucleic acid forms, only HBV RNAs were eliminated by RNase A [11,20,21,22]. However, we concluded that the FISH signals were derived from nuclear cccDNA for the following reasons. First, our probe is complementary to only the gap region of the plus strand in the HBV virion genome (nt 1091–1557) (Figure 1A) [12]. Second, FISH analysis was performed after T5 exonuclease treatment, which cleaves nicked DNA. Finally, the FISH signal was evident only in the nucleus with super-resolution microscopy. 

Peng et al. reported another methodology to define cccDNA’s presence by FISH besides T5 Exonuclease or RNase digestion; drug treatment that can inhibit capsid and reverse transcriptase, hence minimizing the impact of signals from HBV DNA replication intermediates other than cccDNA [21]. This method also resolved the deficiency of the utilization of T5 exonuclease, which might lead to over-digestion and loss of cccDNA [23].

The comparative analysis of parallel qPCR and FISH in both HBV replication and HBV-infected cell lines strengthens the study. Total DNA was extracted, efficiently reducing HBV DNA replicative intermediates, followed by T5 exonuclease digestion, which has been proven ideal, as suggested before the cccDNA quantitative experiment [24].

We examined the dynamics and roles of molecules involved in the regulation of nuclear cccDNA. We previously reported that DOCK11, which belongs to the Zizimin/DOCK family of GEFs [14], may be involved in HBV replication and cccDNA maintenance [10]. Therefore, we evaluated the colocalization of DOCK11 and cccDNA in detail by super-resolution microscopy. However, DOCK11 did not colocalize with cccDNA. cccDNA is a stable episomal form of the viral genome decorated with host histone and nonhistone protein. In recent years, it was clarified that histone modifications on cccDNA regulate viral transcription [17]. Tang et al. [25] showed that transcriptionally inactive cccDNA is not randomly distributed in host nucleus; in the other words, the activation of the cccDNA is apparently associated with its re-localization at specialized areas, including regions close to chromosome 19 (chr.19); these result provided spatial localization of cccDNA.

To investigate how DOCK11 is involved in the interaction between host proteins and cccDNA, among numerous histone and nonhistone proteins, we used H3K4me3, H3K27me3, RNA Pol II, and HP1α as host protein candidates to elucidate DOCK11 function. Methylation of histone H3 lysine 4 (H3K4) is linked to gene activation, while that of H3K27 is commonly associated with gene silencing because it causes chromatin condensation. In addition to the histone proteins, HP1α is a nonhistone chromatin protein. However, various findings demonstrate that nucleosomal HP1α can act as a transcriptional repressor [26], in contrast to RNA Pol II, which is responsible for active transcription. Fluorescence imaging and ChIP results showed that HP1α protein is not included in the cccDNA complex (Figure 5E and Figure 6B) and that H3K27me3 accounted for a small proportion. Meanwhile, H3K4me3a and RNA Pol II C-terminal domain (CTD) significantly contributed to forming the cccDNA structure (Figure 6B). Although DOCK11 did not directly colocalize with H3K4me3, H3K27me3, RNA Pol II CTD, or HP1α (Figure 4F), the ChIP results showed that DOCK11 knockout results in a significantly reduced enrichment of H3K4me3 and RNA Pol II CTD on cccDNA, with HP1α and H3K27me3 remaining unaffected (Figure 6B).

The detailed mechanism of how DOCK11 regulates the H3K4me3 and RNA Pol II CTD that bind to cccDNA is still unclear. The transcriptional procedure is mediated by Ras and Rho family small GTPases to alter gene expression. It has been well documented that an evolutionarily conserved Rho GTPase signaling pathway that targets proteasomes as CTD phosphatases can directly modify the RNA Pol II CTD [27,28]. Indeed, a previous study reported that Rho family GTPases, such as CDC42, can alter the phosphorylation of Ser2 and Ser5 in the CTD of RPB1, the largest subunit of Pol II, thereby affecting gene expression and influencing the size, shape, and number of cells [28]. In addition, CDC42 is involved in the regulation of several transcription factors, including NFκB, SRF, and STAT3. In mammals, the NF-κB/Rel family comprises five proteins: p50, p52, p65 (RelA), Rel-B, and c-Rel. CDC42 can activate the NFκB pathway via a JNK signaling-dependent pathway by a process that involves phosphorylating IκBα and translocating p50/p50 and p50/p65 dimers to the nucleus [26]. The acetylation of p65 has been shown to regulate different histone deacetylases (HDACs), including HDAC-3 and SIRT1 [29]. However, the lysine methylation of promoter-bound STAT3 produces SET9, which is involved in the methylation of H3K4 and is recruited to the newly activated SOCS3 promoter by STAT3, providing docking sites for important functional proteins [30]. These findings suggest that the Rho GTPase CDC42-mediated mechanism is activated by DOCK11. However, more detailed elucidation of the mechanism is required.

In conclusion, we developed a method to more clearly visualize cccDNA and its nuclear binding molecules. Furthermore, we demonstrated the partial co-localization of DOCK11 with H3K4me3 and RNA Pol II, which involves the transcriptional activity of cccDNA. These data provide crucial information for the development of new therapeutics for eliminating cccDNA.

## Figures and Tables

**Figure 1 viruses-15-01178-f001:**
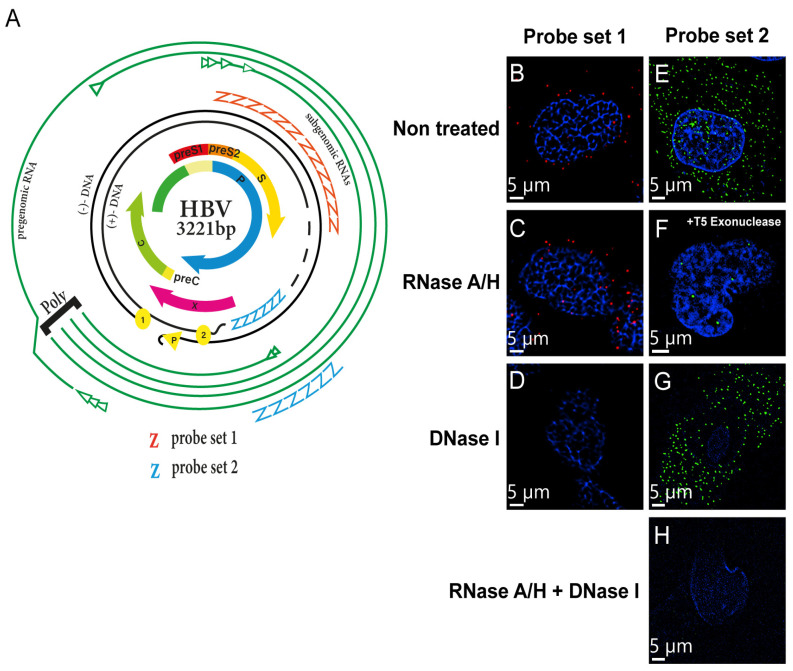
FISH facilitates the determination of key HBV nucleic acids in hepatocytes by confocal microscopy. (**A**) Illustration of the probe set design targeting HBV nucleic acids [12]. Probe set 1 targets the minus strand of HBV DNA. Probe set 2 hybridizes to the sum of pgRNA and subgenomic RNAs and can particularly determine cccDNA when viral RNAs are abolished by RNase A/H. FISH signals were detected in HepAD38 cells to confirm the specificity of FISH signals. The cells were pretreated with RNaseA/H, DNase I, and T5 exonuclease and hybridized with (**B**–**D**) probe set 1 or (**E**–**H**) probe set 2. HBV nucleic acid was analyzed under a Dragonfly confocal microscope with a 60× objective. Scale bar, 5 µm.

**Figure 2 viruses-15-01178-f002:**
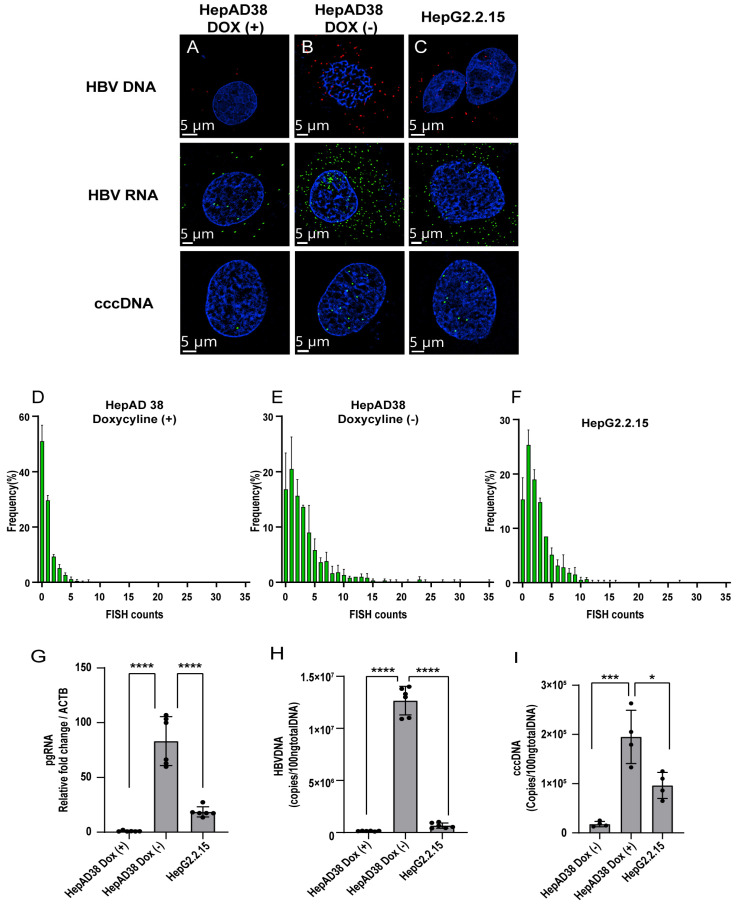
Detection of HBV nucleic acids in HBV-replication cell lines. FISH signals were detected in HepG2.2.15 and HepAD38 cells, pretreated with the indicated reagents, and hybridized with probe sets 1 and 2. HepAD38 cells were maintained in the presence of doxycycline (control) and without the antibiotic (induced). (**A**) Viral replication in HepAD38 cells with doxycycline (1 µg/mL). (**B**) Viral replication in HepAD38 cells was induced by removing doxycycline from the medium. (**C**) Viral replication in HepG2.2.15 cells. Scale bar, 5 µm. FISH counts were derived from a total of 200 nuclei. HepAD38 doxycycline (+) (**D**), HepAD38 doxycycline (−) (**E**), and HepG2.2.15 (**F**). pgRNA (**G**), HBV DNA (**H**), and cccDNA (**I**) from cell lysates were measured by real-time PCR. Representative results from three independent experiments are shown. Statistical differences were assessed by one-way ANOVA. * *p* < 0.05, *** *p* < 0.001, **** *p* < 0.0001.

**Figure 3 viruses-15-01178-f003:**
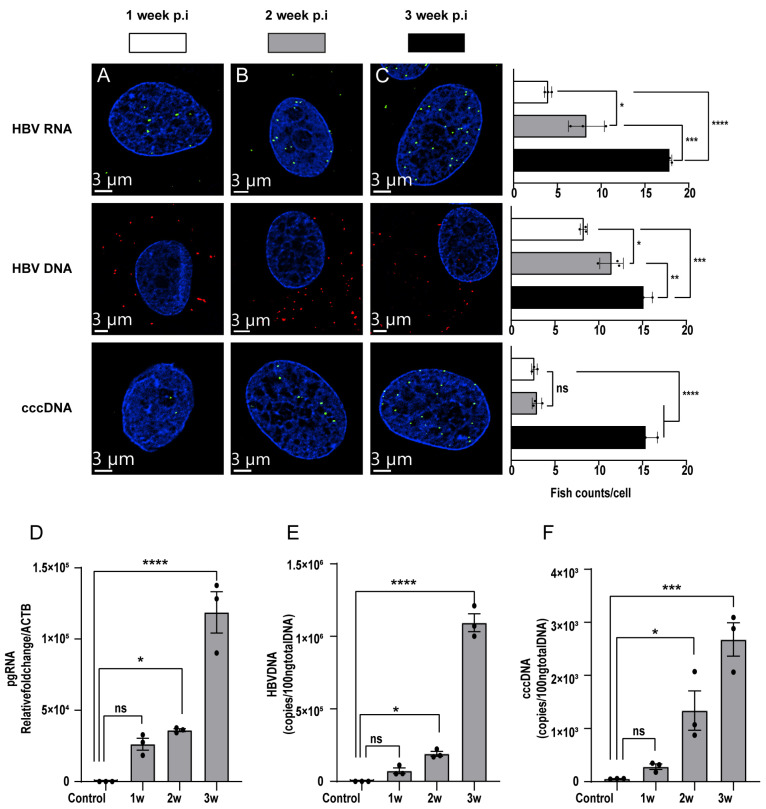
Detection of HBV nucleic acid in infected PXB-cells. PXB-cells infected with HBV genotype C particles 1, 2, and 3 weeks post-infection were plated by passage to 8-well chamber slides for hybridization and DNA and RNA extraction for PCR reactions. Cells were pretreated with the indicated reagents and hybridized with the designated probe sets to detect HBV DNA and HBV RNA (**A**) 1 week after infection, (**B**) 2 weeks after infection, and (**C**) 3 weeks after infection. Images were taken using a Dragonfly confocal microscope using a 60× objective. pgRNA (**D**), HBV DNA (**E**), cccDNA (**F**) from cell lysates from HBV-infected cells, as assessed by real-time PCR. Representative results from three independent experiments are shown. Statistical differences were assessed by one-way ANOVA. * *p* < 0.05; ** *p* < 0.01; *** *p* < 0.001; **** *p* < 0.0001; ns: not significant.

**Figure 4 viruses-15-01178-f004:**
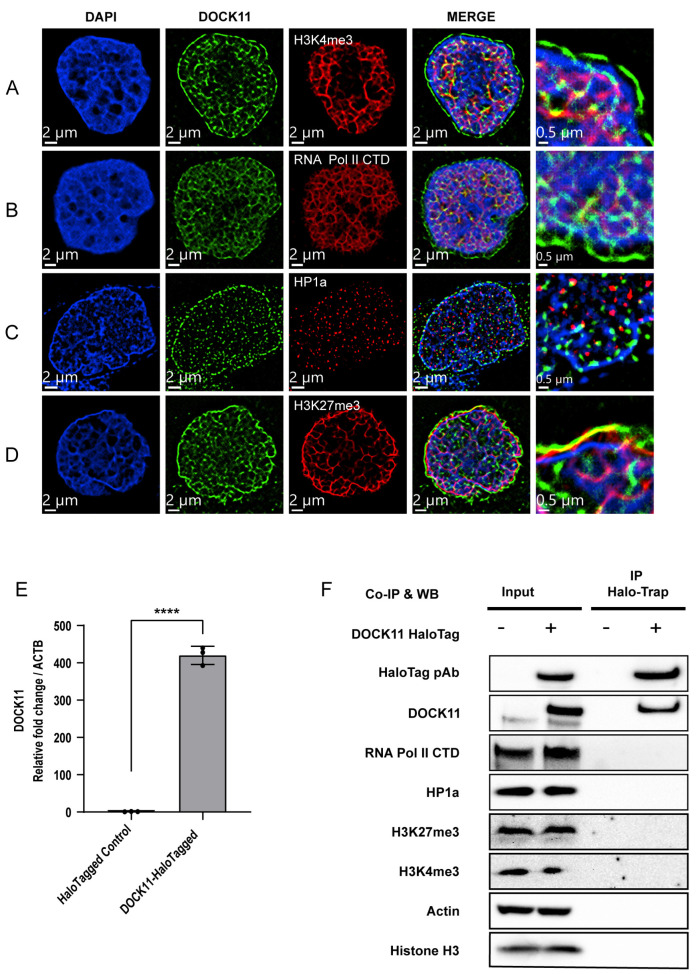
Determination of DOCK11 partners promoting cccDNA replication. (**A**–**D**) Localizations of DOCK11 and several markers. HepAD38 cells were stained with DOCK11 antibody (green) and anti-H3K4me3 antibody (red) (**A**), anti-RNA polymerase II C-terminal domain (CTD) antibody (red) (**B**), anti-HP1α antibody (red) (**C**) or anti-H3K27me3 antibody (red) (**D**). The subcellular localizations of DOCK11, H3K4me3, RNA polymerase II CTD, HP1α, and H3K27me3 were determined by Dragonfly confocal microscopy with a 60× objective. Nuclei were stained with DAPI (blue). Scale bars, 2 μm and 0.5 μm. (**E**,**F**) HepAD38 cells were transiently transfected with Halo-DOCK11 vector or empty vector, and DOCK11 expression was detected by PCR (**E**) and immunoblotting analysis (**F**). Co-immunoprecipitation of DOCK11 and other markers in DOCK11-overexpressing HepAD38 cells by ChromoTek Halo-Trap agarose beads. Representative results from three independent experiments are shown. Statistical differences were assessed by unpaired *t*-test. **** *p* < 0.0001.

**Figure 5 viruses-15-01178-f005:**
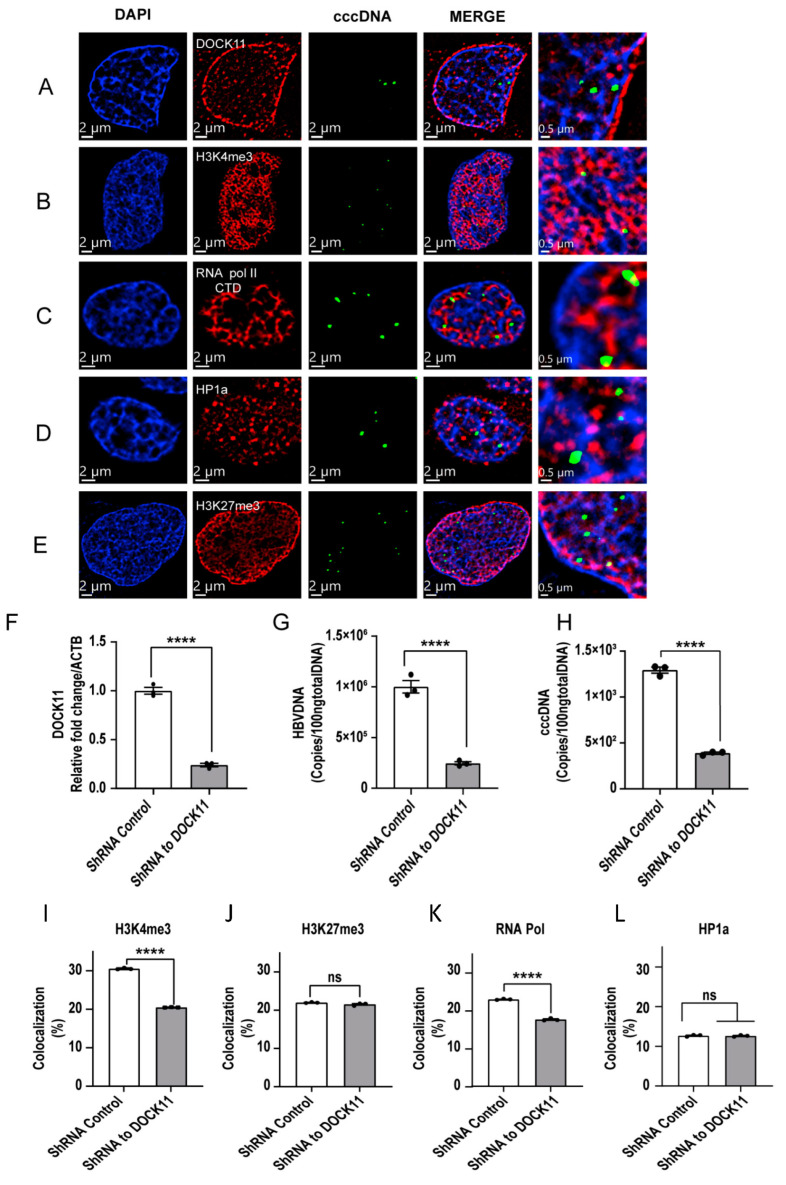
Colocalization of cccDNA with DOCK11 and histone markers, as determined using Dragonfly confocal microscopy. (**A**–**D**) HepAD38 cells were passaged to 8-well chamber slides, pretreated with RNase A/H to eliminate HBV RNAs, and then hybridized to probe set 2 to detect cccDNA (green). Cells were then stained with anti-DOCK11 (red) (**A**), anti-H3K4me3 (red) (**B**), anti-H3K27me3 (red) (**C**), anti-RNA polymerase II CTD (red) (**D**), or anti-HP1α antibody (red) (**E**). Images were obtained using Dragonfly confocal microscopy using a 60× objective. The nucleus was stained with DAPI (blue). Scale bars, 2 μm and 0.5 μm. (**F**–**L**) HepAD38 cells were transiently transfected with DOCK11 shRNA. DOCK11 mRNA (**F**), HBV DNA (**G**), and cccDNA (**H**) levels were detected by RT-PCR. Colocalized signals of cccDNA with H3K4me3 (**I**), H3K27me3 (**J**), RNA polymerase II CTD (**K**), and HP1α (**L**) were captured by a Dragonfly confocal microscope, and then the fluorescence signal counts per cell in each sample were determined using ImageJ software. Representative results from three independent experiments are shown. Statistical differences were assessed by unpaired *t*-test. **** *p* < 0.0001; ns: not significant.

**Figure 6 viruses-15-01178-f006:**
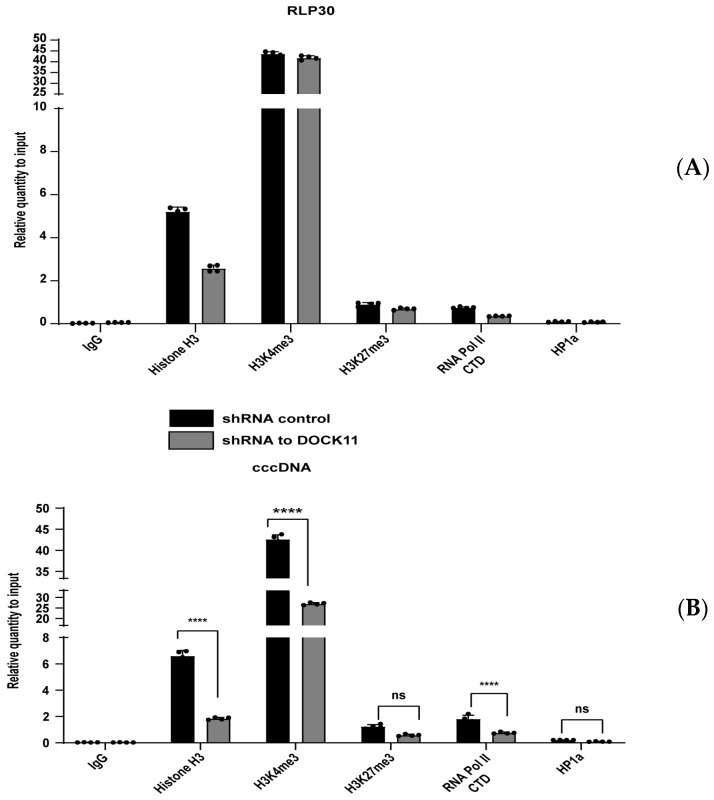
DOCK11 knockdown decreases the levels of cccDNA-associated H3K4me3 and RNA Pol II CTD. HepAD38 cells were treated with DOCK11 shRNA or shRNA control and then cross-linked with 4% formaldehyde solution, sonicated, and immunoprecipitated with the indicated antibodies. The levels of H3K4me3, H3K27me3, RNA pol II CTD, and HP1α associated with the RLP30 promoter (**A**) or HBV cccDNA (**B**) upon DOCK11 knockdown were determined by ChIP-qPCR. Representative results from three independent experiments are shown. Statistical differences were assessed by two-way ANOVA. **** *p* < 0.0001; ns: not significant.

**Table 1 viruses-15-01178-t001:** Distribution of cccDNA in HBV-inducible cell lines.

(cccDNA/cell)		HepAD38 (Dox^+^)	HepAD38 (Dox^−^)	HepG2.2.15
**FISH counts**	Average	1.51	5.59	3.9
	Median	4	11	11.5
	Min./Max	0/8	0/31	0/32
**PCR measurement**	Average	1.07	11.1	5.79

The table summarizes the cccDNA-FISH counts in HepAD38 and HepG2.2.15 cells. The average and median are the average and median counts from 200 nuclei derived from HepAD38 cells with and without doxycycline and from HepG2.2.15 cells. Min./Max., minimum/maximum (range of FISH counts). The cccDNA number was measured by real-time PCR.

## Data Availability

The datasets supporting the conclusions of this study are included in the article.

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
