# Peer review of "Super-Resolution Microscopy Analysis of Hepatitis B Viral cccDNA and Host Factors"

_viruses, 2023, doi:10.3390/v15051178_

Round 1

Reviewer 1 Report

The authors described a new method to visualize more clearly cccDNA and associated molecular host factors by means of super-resolution microscopy. Please find below some questions in order to improve in their manuscript.

1)    Could the authors check the sentence at lines 283-286 for contradiction (inversion between H3K27me3 and H3K4me3) with sentences at lines 53-56 and lines 402-403 ?

2)    Could the authors further discuss how their method allow to visualize more clearly cccDNA in regards to ICE-HBV protocol (Ref. 11) or PMIDs: 29046450 34192543 34674513 or  36475867 ?

3)    Could the authors discuss how their quantification of cccDNA in cell nuclei compare to PMID: 36707234 ?

4)    Could the authors check the use of “interaction” at line 435 in the light of the sentence at lines 297-299/410-413, and clarify the sue of colocalize/associate/bind/interact throughout the manuscript ?

5)    Could the authors comment on the use of HBV of different genotypes (C and D) depending on experiments carried out.

6)    Could the authors change the second “probe set 1” in “probe set 2” in Figure 1 legend (line 207) ?

7)    Could the authors check the Figure 3 legend for labels D E F (line 262) ?

8)    Could the authors check the Figure 4 legend for labels A-D with corresponding H3K4me3/H3K27me3/RNA polymerase II/HP1alpha (lines 292-295 & 302-304) ?

Reviewer 2 Report

The authors use SR-Microscopy to visualize HBV-RNA and HBV-DNA forms in stable HBV-expressing cells as well as infected hepatocytes. The authors use this technology to elucidate the effect of DOCK11 on HBV cccDNA maintenance. 

General comments:

1   While I believe the observation by the authors is real and that FISH technology can be use for the purpose of HBV DNA measurement, effects during the FISH staining methods between the different samples can lead to variable results. By using 200 nuclei the authors tried to get an average of DNA distribution per sample, however the total number of Samples analyzed is still N=1 in all the experiments. This is comparable with a FACS analysis of a single well. It will give you an average of a huge number of cells, however, the total number of Samples analyzed will be N=1. Therefore, FACS analysis usually uses N>3 samples and combine the results.  I would  therefor suggest to use 3 independent experiments including independent stainings to strengthen the papers results.

2   The authors count the number of Fish counts per cell, however, they do not show how a cell can be detected? Did the authors use WGA or any other membrane staining to distinguish on cell from another?

     Specific comments:

Line 37: There are  different forms of HBV DNA in the nucleus, rcDNA, cccDNA and integrated HBV DNA. Please rephrase line37 “Because of its supercoiled nature, HBV cccDNA in the nucleus of infected” hepatocytes.

Line 53: Please include a short explanation about the KM cell line in the introduction.

Line 80: There is no information about the HBV particles used for this publication online available as referenced #PPC-BC; 80 PhoenixBio. Please include more information on how they were purified and from which cells.

Line 167: please remove 3.1.1. from the titel

Line183-185: HepAD38 cells harbor HBV genome integrations. Will the DNAse step also remove cellular genome that there is no signal left?

Figure 2: “Detection of HBV nucleic acids in HBV-inducible cell lines.” only HepAD38 is inducible not HepG2.2.15.

Figure 2A: As already mentioned HepAD38 cells habor HBV in their Genome. Please explain why you don’t see this with this method?

Figure 2 D-F: In order to compare the frequencies. I would suggest to keep the axis similar.

Line 229: How were HepG2.2.15 induced?

Figure 3: The authors do not explain the observations, HBV RNA is is now in the nucleus and not in the cytoplasm and on the other hand, that HBV DNA is now in the cytoplasm and not in the nucleus.

Figure 3: the authors should change the graph on the right (fish counts) that 1 week is top and 3 weeks is bottom.

Figure 3: Other papers suggest that cccDNA stays at a plateau after 1 Week can the authors explain why they see this drastic increase.

 Line 428-431: The authors did not demonstrate that the interaction of DOCK11 with H3K4me3 and RNA Pol II regulate the transcriptional activity of cccDNA. They only showed a partial colocalization and an effet of DOCK11 knowdown on the expression levels of these proteins. There could be an other factor involved in this mechanism which is responsible for the effect. This part of the conclusion needs to be revised.

Round 2

Reviewer 2 Report

The Manuscript has been improved by the authors corrections. While the authors replied to all of the comments, I would like to respond to the following comment:

Reviewer: Figure 2A: As already mentioned HepAD38 cells habor HBV in their Genome. Please explain why you don’t see this with this method?

Author: The HepAD38 cell line harbors the whole HBV genome, that regulates expression of pgRNA. In our experiment, we did HBV-FISH only, so we couldn’t observed whether HBV genome integreted into chromosome or not. To define the interaction of HBV genome and host, we need to do further experiment to analysis which chromosome location that HBV intergated, as well as label the chromosome. 

Reviewer: I still don't understand why your HBV-FISH Probe is not capable to bind HBV-DNA in the Genome if it is capable to bind to cccDNA. Please explain this in the text or discussion. 

--

Also I would suggest to include the Sequences of the Probes used in this publication in the supplement.
